# High Temperature Electrical Properties of Co-Substituted La_4_BaCu_5_O_13+δ_ Thin Films Fabricated by Sputtering Method

**DOI:** 10.3390/ma14102685

**Published:** 2021-05-20

**Authors:** Woosuck Shin, Akihiro Tsuruta, Toshio Itoh, Takafumi Akamatsu, Ichiro Terasaki

**Affiliations:** 1National Institute of Advanced Industrial Science and Technology (AIST), Shimo-Shidami, Moriyama-ku, Nagoya 463-8560, Japan; a.tsuruta@aist.go.jp (A.T.); itoh-toshio@aist.go.jp (T.I.); t-akamatsu@aist.go.jp (T.A.); 2Department of Physics, Nagoya University, Furocho, Chikusa-ku, Nagoya 464-8602, Japan; terra@nagoya-u.jp

**Keywords:** sputtering thin film, perovskite oxide, cuprate, Seebeck coefficient

## Abstract

The high-temperature conductivity of the perovskite oxides of a La_4_BaCu_5_O_13+*δ*_ (LBCO) thin film prepared by RF sputtering deposition and thermal annealing has been studied. While the bulk LBCO compound was metallic, the LBCO film deposited on a Si substrate by sputtering and a post annealing process showed semiconductor-like conduction, which is considered to be due to the defects and poor grain connectivity in the LBCO film on the Si substrate. The LBCO film deposited on a SrTiO_3_ substrate was of high film quality and showed metallic conduction. When the cation site Cu was substituted by Co, the electrical conductivity of the LBCO film increased further and its temperature dependence became smaller. The transport properties of LBCO films are investigated to understand its carrier generation mechanism.

## 1. Introduction

Perovskite-related copper oxides crystallize in a variety of structures, which provide many interesting material properties, especially electrical transports. As it is well known, high-*T*_c_ superconductivity is realized in layered cuprates with two-dimensional CuO_2_ planes. The superconductivity was also achieved in quasi-one dimensional cuprates with a Cu_2_O_3_ ladder such as (Sr,Ca)_14_Cu_24_O_41_ [1]. Michel et al. have investigated a La_4_BaCu_5_O_13+δ_ system and found that these perovskite-related copper oxides are highly metallic in conduction with a resistivity of 1.5 mΩcm at 673 K [2], and Murayama et al. have substituted Sr for Ba to demonstrate metallic conduction but no superconductivity [3]. The conducting oxides are used in a vast range of applications, such as sensors [4], solid oxide fuel cells (SOFC) [5], energy devices [6], and electrical leads, and are indispensable for all of these applications. Among them, the cuprates of various elemental compositions are promising materials because of their natural abundance and high-temperature stability, which are superior to metals’. Electrical properties from room to high temperatures over 900 K, for example, SOFC operating temperature, and lower resistivity around mΩcm are important for various applications.

As shown in Figure 1, Michel et al. reported the crystal structure of La_4_BaCu_5_O_13+δ_ (LBCO) as tetragonal, space group P4/m, which is related to the cubic perovskite sub-cell, combining one CuO_6_ octahedral plus four corner-sharing CuO_5_ pyramids with protruding red spheres of Cu, and a one-dimensional CuO_6_ octahedral chain to the *c*-axis [7]. Each octahedron shares four corners with four pyramids and two corners with two other octahedra. The framework exhibits one perovskite-like tunnel and two hexagonal tunnels per cell.

In the LBCO, the Cu ion of octahedral coordination in particular were expected to be substituted by the other transition metal ions, such as Ni, Co, or Fe trivalent ions [9], and Ni^3+^ and Co^3+^ were reported to partially occupy the one-fold octahedral and four-fold square pyramidal sites [10]. Increased oxygen content was observed with Ni- and Co-substitution, leading to an increase in unit-cell volume. The undoped and Ni-doped phases are metallic Pauli paramagnets; the Co- and Fe-doped materials show a metal-to-insulator transition with increasing the Ni(Co) content *x* and are weakly paramagnetic [11]. Mori et al. reported that the LBCO bulk samples showed metallic temperature dependence with a low resistivity of less than 2 mΩcm at 800 K, indicating that the metallic conduction is robust against impurities of up to 7% Co-doping for the Cu site [12]. They discussed that the robust good metallic conduction is responsible for two components of the electron and hole conduction, and that the impurity potential seems to be screened by the high electron density.

Further study on the electrical anisotropy of this oxide material has been reported by the fabrication of thin films on a SrTiO_3_(100) single crystal substrate by pulsed laser deposition (PLD), and has shown that the temperature dependence of the resistivity along the *ab*-plane is smaller than that along the *c*-axis in the thin-film study [13]. The epitaxial thin films of Co-doped LBCO on a SrTiO_3_ substrate have been also fabricated by PLD by Tsuruta et al., who measured the resistivity and Seebeck coefficients for a wide temperature range from 10 to 1000 K, and suggested a change of carrier scattering with temperature from the impurity scattering of Co to phonon scattering at 700 K [14]. However, other dependences of the electric properties on the Co substitution, the film thickness, or quality, such as the lack of homogeneous film; different substrates, including glass; and oxygen deficiency, especially at temperatures higher than room temperature RT, have not been investigated in detail.

In the present study, we prepared a La_4_BaCu_5_O_13+δ_ thin film through the sputtering method, which is a more preferable fabrication process for industrial applications. Sputtering is used extensively in a wide-range of applications to deposit thin films of various materials from metals to oxide insulators. Several important advantages of sputter deposition are that the sputter deposited films have a composition close to that of the source or target material, the reproducibility of the process, and the scalability for mass production over other thin film processes. Advanced processes, such as epitaxial growth, are also possible under the control of the substrate material and temperature, and after thermal annealing treatment.

In this study, a control of the film composition, Co-doping for the La_4_BaCu_5−*x*_Co*_x_*O_13+*δ*_ film, was carried out by oxide chips on the sintered target of La_4_BaCu_5_O_13+δ_ during the sputtering, and the effects of various substrates of SiO/Si vs. SrTiO_3_ single crystal substrates were investigated. The effects of post annealing temperature, atmosphere, and Co substitution on the electrical properties (Seebeck coefficient and conductivity) were explored and discussed in regard to the microstructure of the films and the two-carrier conduction model.

## 2. Materials and Methods

### 2.1. Thin Film LBCO Preparation

The starting materials of LaO, Ba(OH)_2_, and CuO of the stoichiometric mixture were grounded and compacted to prepare the sputtering target (La_4_BaCu_5_O_13.5,_ diameter 100 mm, supplied by Toshima Manufacturing Co., Ltd, Saitama, Japan). Considering the ion composition of the target La_4_BaCu_5_O_13.5_ and the chip Co_3_O_4_, there was approximately two times the Co of the chip than the Cu of the target, the Co substitution for Cu in the film was estimated to be 2.5% in the co-sputtering with a single chip, corresponding with a nominal composition of Cu_4.875_Co_0.125_. Figure 2 shows the configuration of the chips on the target.

Three different substrates of a SiO/Si(100) wafer with a thickness of 0.5 mm, quartz glass with a thickness of 1.0 mm, and a SrTiO_3_ (100) single crystal with a thickness of 1.0 mm were used. During the sputtering deposition, the substrate temperature was 200 °C, the working pressure was 1.7 × 10^−1^ Pa with an argon flow rate of 20 cc/min, and the RF power was 100 W. The deposition rate was checked by a surface profilometer (P-17 stylus profiler KLA-Tencor Co., Milpitas, CA, USA) to be 6 nm/min, and the film thickness was controlled to 500 nm. After the deposition, a post process of thermal annealing of the LBCO film was carried out to enhance the crystallinity and to induce high mobility of the sputtered film, with temperatures of 600, 800, and 1000 °C and an annealing time of 4 and 12 h in an air atmosphere.

### 2.2. Morphology Characterization and Electrical Property Measurements of Thin Films

The phase and crystallinity of the prepared samples were investigated using X-ray diffraction (XRD) using two standard diffractometers, one for the films on SiO/Si substrate (RINT 2100 V/PC, Rigaku, Tokyo, Japan) and the other with a Ni monochromator for the films on SrTiO_3_ (SmartLab, Rigaku, Tokyo, Japan) with CuKα radiation in the 2*θ*-*θ* scan mode. The morphology of the bulk samples was observed using a field emission scanning electron microscope (FE-SEM) (JSM-6335FM, JEOL, Tokyo, Japan). In addition, the surface morphology of the films was observed by an atomic force microscope with a contact mode using Si_3_N_4_ cantilever tips (Seiko Instruments Inc., SPI-3800N Probe station, Tokyo, Japan).

A commercial electrical conductivity and Seebeck coefficient measurement system (RZ20001i, Ozawa Science Co., Nagoya, Japan) with an option of thin film measurements with four-point pressure-contact electrodes was used for measurements of the conductivity, σ, and Seebeck coefficient, α, of the thin-film LBCO samples from 50 to 600 °C. One side of the sample was cooled down by air flow to induce a temperature difference in the sample, as described in detail in a previous report [15]. Hall carrier density and charge carrier mobility of the film was measured using a commercial Hall measurement system (Resitest8300, Toyo Co., Tokyo, Japan) by the van der Pauw method with this four-point electrode system and a magnetic field of 0.75 T.

## 3. Results

### 3.1. XRD and Surface Structure of LBCO Films after the Thermal Annealing

Figure 3 shows the XRD patterns of LBCO thin films after the post annealing at 600, 800, and 1000 °C for 4 h. The peaks assigned to LBCO on the SiO/Si substrate were checked to evaluate the crystal phase of the thin films, depending on the thermal annealing temperature.

The as-deposited film was amorphous, but the LBCO peaks appeared after the thermal annealing at 600 °C, as shown in (1) of Figure 3a. The film annealed at 800 °C for 4 h showed relatively better crystal structure, and its diffraction peak corresponding to the (121) planes with 2θ = 33° was a prominent peak, as shown in the peak pattern of (2) of Figure 3a.

Regarding the identification of the heterogeneous phases at 1000 °C, as indexed by other phases, we investigated the peaks using Pearson’s crystal data, but did not hit with the likely oxides of the La-Cu/Cu-Ba system. For the possibility of Si diffusion forming a compound, the related peaks at 31° at 800 °C seemed to be 300 planes and 201 planes, but the related peaks (600 and 402) were originally weak, and the peak shape was sharper than the others, so it was difficult to judge.

After high-temperature thermal annealing at 1000 °C for 4 h, thermal decomposition occurred so that many peaks of the other phase also appeared, and the LBCO phase was reduced, as shown in (3) of Figure 3a. The crystallinity of the film was no worse than that of the film on the Si substrate, but the relative peak intensity of the single crystal STO substrate was very strong, thus that the peak of LBCO looked relatively weak. From this result, a moderate temperature of 800 °C for 4 h was chosen for the post annealing process of the LBCO film on SrTiO_3_ crystal.

As shown in Figure 3b, the LBCO films on the SrTiO_3_ single crystal substrate were in single phase and of better crystallinity, and the peaks to specific crystal planes became distinct. For the lattice mismatching, without crystal rotation, the lattice misfit between the a-axis of LBCO (≈8.65 Å) and the length of a two-unit cell of SrTiO_3_ (7.82 Å) would be 10.6%, but with 26.6° rotation, lattice misfit became −1.1%, as discussed in a previous report [14]. However, there is no clear evidence of grain orientation, despite the lattice parameters of (121) for LBCO and (101) for SrTiO_3_ being identical, which suggests that the grains of the deposited film were rather randomly oriented. The invariance of the *c*-axis length with Co-substitution has been reported in bulk [12]. No peak for Co_3_O_4_ (PDF no. 00-042-1467) was detected in the Co-substituted LBCO film. The exact composition is unknown, but we regarded the substitution as occurring after high-temperature annealing, following the above discussion.

### 3.2. Electrical Properties of LBCO Films on SiO/Si and SrTiO_3_ Substrates

Figure 4 shows the thermoelectric properties of the LBCO films, the Seebeck coefficient, α, and the conductivity, σ, measured at temperatures from 50 to 480 °C in air. The temperature dependencies of the σ measured clearly indicate that the LBCO films of high conductivity were metallic, and those of poor electrical conduction were semiconductor-like.

These results seem to be due to the grain boundaries in the films. The current path of grain boundaries has a significant effect on the electrical conductivity of LBCO films, and the conductivity of the 0.5 μm film on SiO/Si was extremely low, with the α being estimated to be negative around −500 μV/K at high temperatures. The measurement of the high Seebeck coefficient over the mV range is worth being checked again. The middle and high temperature range Seebeck coefficients were confirmed by cyclic measurement by cooling from a high temperature to a low temperature, where the resistance of the sample became low and the Seebeck measurement became highly reliable and repeatable at the high temperature.

A thicker film of 1 μm was prepared and a moderate thermal annealing condition at a lower temperature of 600 °C for 12 h was applied to improve the conductivity of the film, resulting in better electrical conduction. This thicker film showed a positive α value around 80 μV/K. Neither the large negative value nor the change of the sign of the α can be explained by the effect of grain boundaries.

The reported conductivity value of dense LBCO bulk oxide sintered at high temperature was as high as 590 S/cm at around room temperature [16]. The conductivity of the LBCO film on the Si substrate in this study was much lower than that of reported, and furthermore, their temperature dependence was semiconductor-like.

However, the thermoelectric properties changed drastically by changing the substrate. The LBCO film on the SrTiO_3_ substrate showed similar thermoelectric values to the reported bulk properties. Comparing the thick film LBCO on SiO/Si and the thin film LBCO on SrTiO_3_, their Seebeck coefficients were similar, 79 and 42 μV/K, respectively, at 250 °C, but the temperature dependence of the conductivity changed from semiconductor-like to metallic. The temperature coefficient of resistance, TCR, of the LBCO film became small as the quality of the film was improved. The α of the LBCO thin film on SrTiO_3_ varied slightly with temperature, ranging from 38 to 48 μV/K.

Figure 5 shows the thermoelectric properties of the LBCO films of different Co-substitution on SrTiO_3_ measured from 50 to 480 °C in air. The temperature dependency of the σ of the LBCO on SrTiO_3_ was very weak, and the σ of the Co-substituted thin films had much higher conductivity than that of the non-substituted ones. The α of the Co-substituted thin films was smaller than the non-substituted LBCO films, and their behavior become almost the same in the high-temperature region and slightly increased with temperature, ranging from 8 to 18 μV/K.

## 4. Discussion

### 4.1. Microstructure of the LBCO Films on Different Substrates

The surface morphologies of the LBCO films were observed for a better understanding of the relation between the grain growth, thermal annealing, and the different substrates. Figure 6 shows their microscopic AFM and FESEM images. The AFM images of the surface of the as-deposited LBCO films on the SiO/Si substrate and SrTiO_3_ look similar, as shown in Figure 6a–c. The grain size was around several hundred nanometers for the films on the SiO/Si substrate, and it became smaller than one hundred nanometers for the film on SrTiO_3_. The agglomerate grain structure of 200–300-nm-wide irregular shapes that was formed on the surface of the films on SiO/Si shows hillocks. Due to these hillocks, it was difficult to analyze the surface roughness quantitively on a nanometer level. In the case of Co-substituted LBCO film, no distinct change was found, but the agglomerate grain structure was unclear for this sample.

However, after high-temperature thermal annealing, the surface of the film on SiO/Si changed from a mirror surface to a rough surface, and this was confirmed by the peel off of the film, as shown by the FESEM observation in Figure 6d. In the case of a thicker film—1μm LBCO film on SiO/Si—it was significantly improved, and the peeled off area was reduced. Figure 6e intentionally shows the peeled off area to observe the film thickness. The SEM observation of the LBCO film on SrTiO_3_ showed just a smooth surface, and the details are shown in Figure 6c.

From these microstructure observations, the difference in the electrical conductivity can be explained. These results seem to be due to defects such as cracks in the films. The films on the SiO_2_/Si showed poor grain connectivity, which induced low charge mobility and low carrier concentration. As the cracks had a significant effect on the electrical conductivity of LBCO films, the conductivity of the 0.5 μm film on SiO/Si was extremely low.

In Table 1, the transport properties of typical metals and LBCO materials of bulk and films are compared. The bulk, PLD film, and sputter film of LBCO on SrTiO_3_ showed low resistivity around 0.5 mΩ, but was still higher compared to those of metals at 0.01 to 0.12 mΩ. This is due to the fundamental limit of oxide materials. The orientation of the SrTiO_3_ or Si substrate was an important factor regarding the film growth, and also effected the resistivity of the film. In the case of Si, as the film grew on the SiO_2_ surface, the lattice matching was not critical. However, in the case of the SrTiO_3_ crystal substrate, lattice matching would lead to this low resistivity.

As listed in Table 1, the temperature coefficient of resistance, TCR, of the metals Pt or Cu was positive, but the that of constantan alloy was negative. Most of the LBCO samples showed positive TCR in this study, but the that of sputter-deposited film and that reported for the PLD film showed negative TCR. The TCR of previously reported LBCO-related oxides of La_1−x_A_x_BaCuO_2.5_ (A = Ca, Ba, Sr) [19], La_4_BaCu_3_Co_2_O_13+δ_ [20] and La_4_BaCu_3−x_Mn_x_O_13+δ_ composition [21] for the application of fuel-cell electrodes showed similar behaviors of negative TCR up to 400 °C and positive TCR at high temperatures, which was related to the loss of oxygen. The very small TCR of the LBCO in this study is characteristic, which may be promising property for the application of resistant materials.

The signs of α of the LBCO were positive from 4 to 13 μV/K, and the major carrier was considered to be a hole. In the case of the sputter film of LBCO on SiO/Si, however, the change of the sign of α was not simple. The thin film showed negative −510 μV/K at high temperatures. The thick conducting film showed a positive α value around 50 μV/K.

### 4.2. Change of Seebeck Coefficient and Discussion on the Two Charge Carriers’ Model

The thick LBCO film sample deposited on Si or SrTiO_3_ exhibited p-type, hole carrier conduction. However, thin LBCO films with a lot of cracks on the Si substrate showed n-type and semiconductor-like conduction. This inversion of charge carrier type can be explained by a two-carrier model.

The fact that oxygen-deficient LBCO is thermodynamically stable in a reduced high-temperature atmosphere [22,23], resulted in thermal annealing producing a higher concentration of oxygen vacancies and electrons, which could involve a systematic reduction in the coordination of the octahedral Cu groups to square pyramidal; square planar; and, finally, linear coordination [23].

During the annealing, the film became crystallized, but the hole in the LBCO film was reduced, and also induced a change in α from positive to negative, indicating a change in the major carrier type from p-type to n-type. At high temperatures in air, oxygen vacancies and electrons became increased, so that the conductivity increased with temperature. To understand the carrier type, we measured the Hall resistivity, and the result obtained was negative, showing that the carrier type was electron, which is the opposite result to the sign of the Seebeck coefficient. Assuming a single carrier model, the temperature dependence of the carrier concentration and the mobility of La_4_BaCu_5_O_13+δ_ thin film on the substrate of SrTiO_3_ was plotted as shown in Figure 7.

In the two-carrier model, the Seebeck coefficient, α is described by
$$\alpha =\frac{\sigma_{h}\alpha_{h}+\sigma_{e}\alpha_{e}}{\sigma_{h}+\sigma_{e}}$$
where σ_h_ and σ_e_ are the conductivities for holes and electrons, respectively. Hence, the magnitude of α is determined by the balance between the conductivity and the Seebeck coefficient of each carrier. As Mori et al. have pointed out that electrons are majority carriers, which is consistent with the negative Hall resistivity, they dominate the Hall measurement carrier density in LBCO [12]. On the other hand, considering that the fraction of the α_e_ is as small as 4 μV/K at room temperature [10], the large positive Seebeck coefficient of the LBCO can be regarded as the contribution or concentration of holes. Mori et al. have shown a quantitative analysis of the two-carrier model by a varying magnetic field up to 7 T, and their parameters are comparable to the reported values. Though the analysis was not discussed in detail due to the small Hall voltage at low magnetic fields, we can also understand the origin of the positive Seebeck coefficients of the LBCO films on the substrate of SrTiO_3_. For the LBCO films on Si, as the mobility and concentration of the hole carrier was extremely reduced due to cracks and oxygen detects, the contribution of α_e_ became prominent and the sign of the Seebeck coefficient was negative.

## 5. Conclusions

We investigated the high-temperature thermoelectric properties of LBCO thin films prepared by RF sputtering deposition with different process parameters of the substrate, and Co-substitution for the Cu ion site. While the bulk LBCO compound was metallic, the LBCO film deposited on a silicon substrate by sputtering and a post annealing process showed semiconductor-like conduction, which was considered to be due to the defects and grains in the LBC film on the Si substrate. This LBCO film of poor conductivity showed a high temperature dependence to conductivity, and a negative sign of Seebeck voltage. From these results, we can conclude that the quality of the film significantly modifies its thermoelectric properties, not only the conductance but also the Seebeck coefficient.

In contrast, the LBCO film deposited on a SrTiO_3_ substrate was of high quality and conductivity and showed metallic conduction. When the Cu cation was substituted by Co, the electrical conductivity of the LBCO film increased and its temperature dependence became smaller. In this study, it was proven that LBCO sputter deposition processes are adaptable from simple oxide target materials, chips of substitution materials, and the effect of the thermal process and substrates. The results of conductivity, the Hall effect, and the thermopower of LBCO films up to high temperatures were discussed and understood by the microstructure of the films and the two-carrier model of the small fraction of the hole component of high mobility.

## Figures and Tables

**Figure 1 materials-14-02685-f001:**
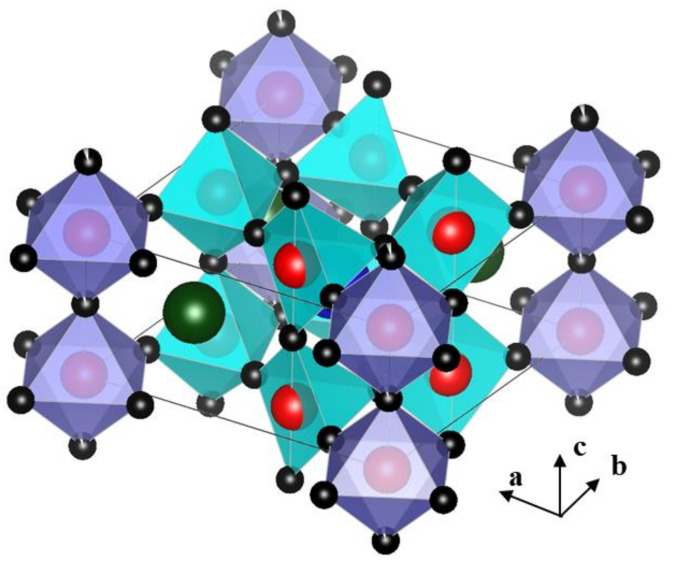
Crystal structure of La_4_BaCu_5_O_13+δ_. (The figure was drawn using VESTA [8]. Red is Cu; blue is Ba, which is shaded in the pyramid; green is La; and black is O).

**Figure 2 materials-14-02685-f002:**
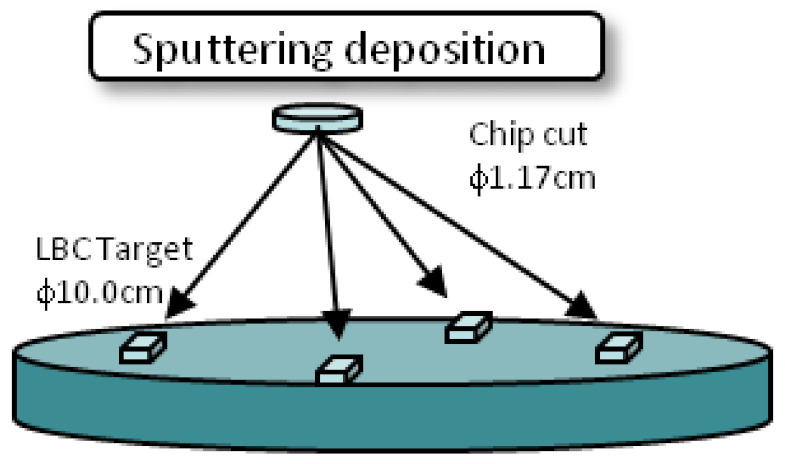
The chips of Co_3_O_4_ were placed on the surface of the LBCO sputtering target of 100 mm in diameter.

**Figure 3 materials-14-02685-f003:**
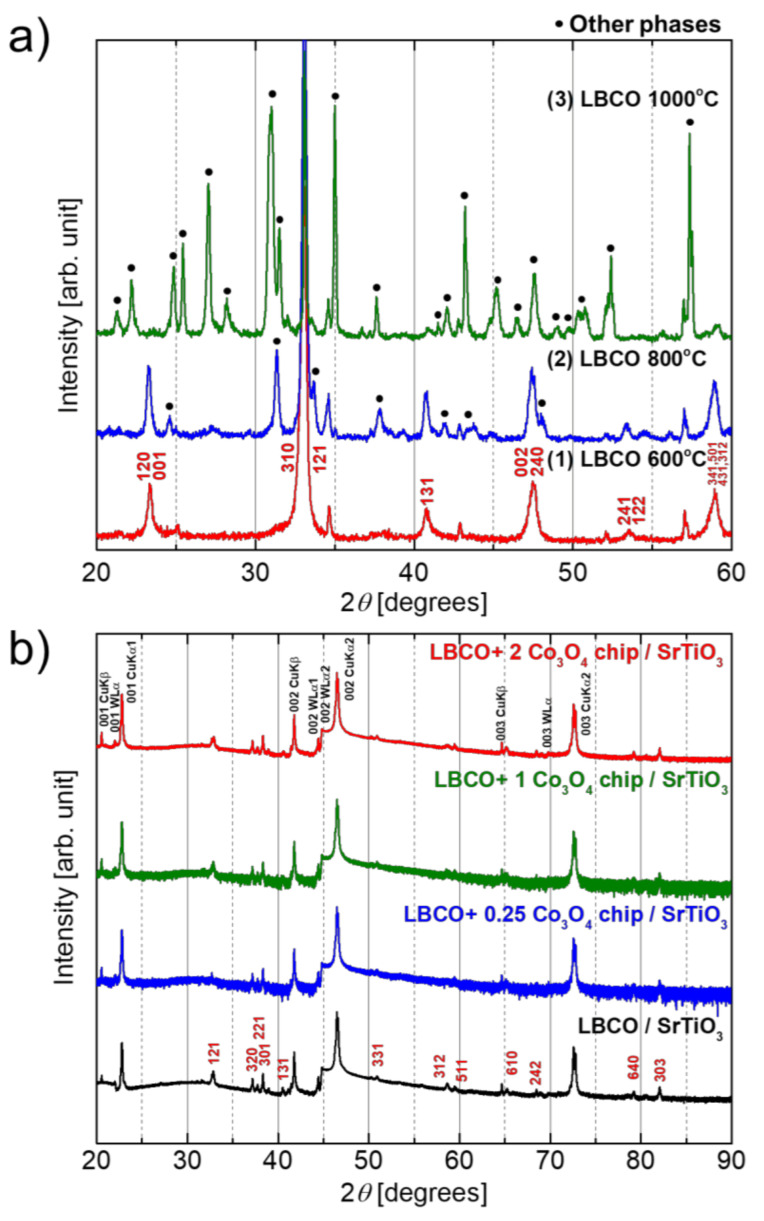
XRD pattern of the LBCO and Co-substituted LBCO films, (**a**) with different thermal annealing temperatures on the SiO/Si substrate, and (**b**) the LBCO films (the peaks of red index) compared with the SrTiO_3_ single crystal substrate (the peaks of black index) with the different concentrations of Co-substitution.

**Figure 4 materials-14-02685-f004:**
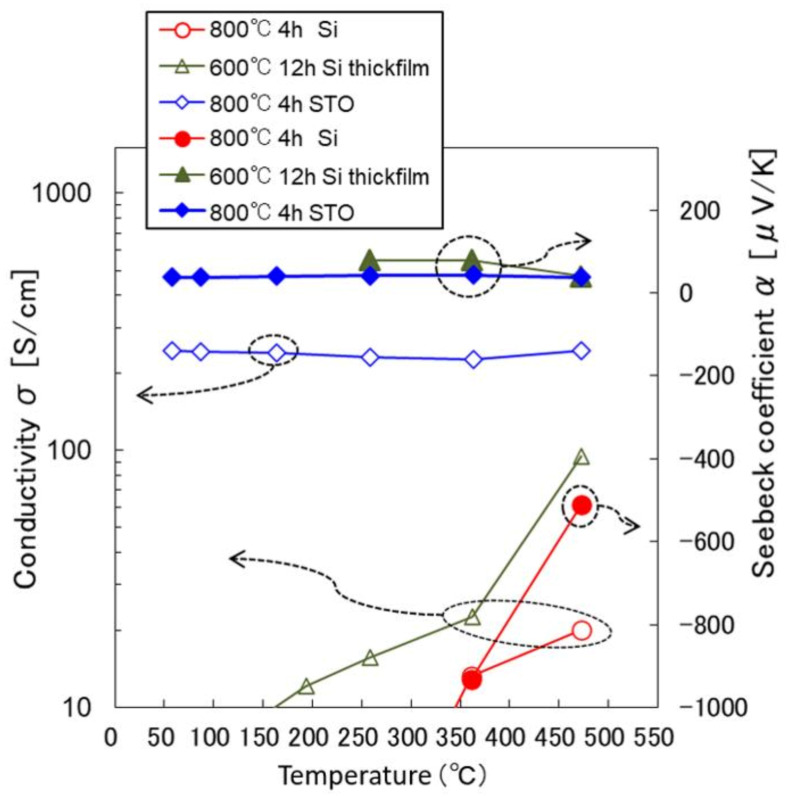
High-temperature thermoelectric properties of the LBCO films.

**Figure 5 materials-14-02685-f005:**
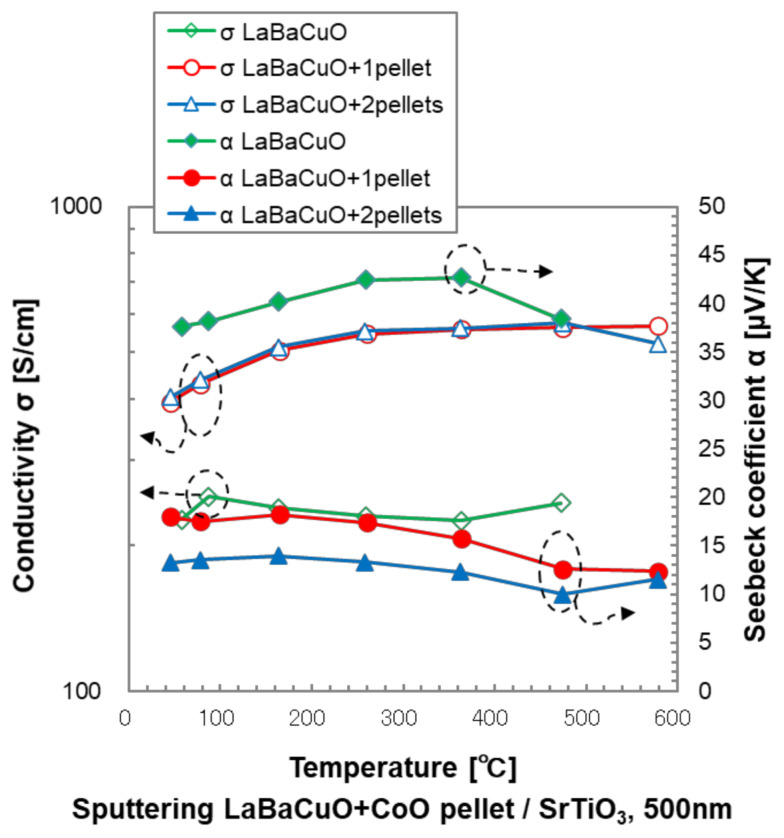
High-temperature thermoelectric properties of the Co-substituted LBCO films.

**Figure 6 materials-14-02685-f006:**
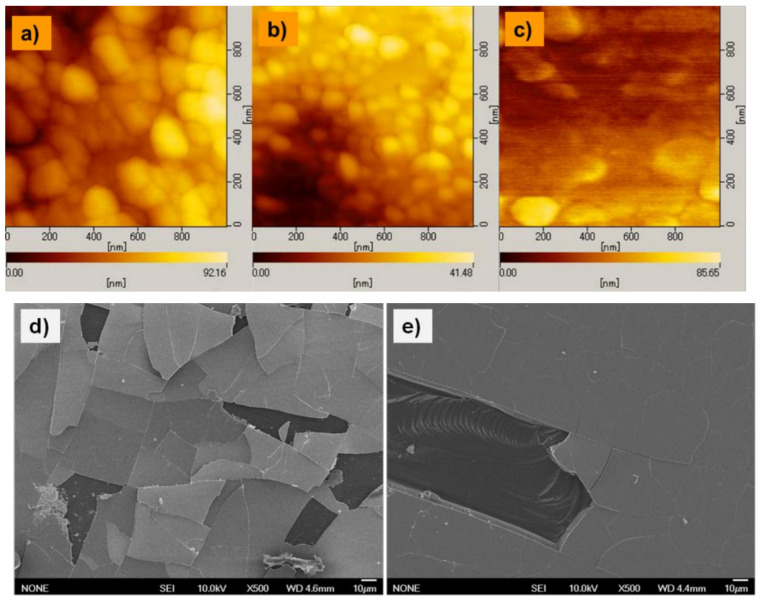
AFM and FESEM micrographs showing the surface morphology of the prepared LBCO films on (**a**)SiO/Si, (**b**) SrTiO_3_, and (**c**) Co-substituted film on SrTiO_3_. SEM observation of the LBCO films on Si wafers showed a high crack concentration (**d**) in the film with a thickness of 0.5 μm, and (**e**) greatly reduced cracking in the thicker, 1 μm, film. The SEM observation of the film on SrTiO_3_ showed just a smooth surface and is not shown here.

**Figure 7 materials-14-02685-f007:**
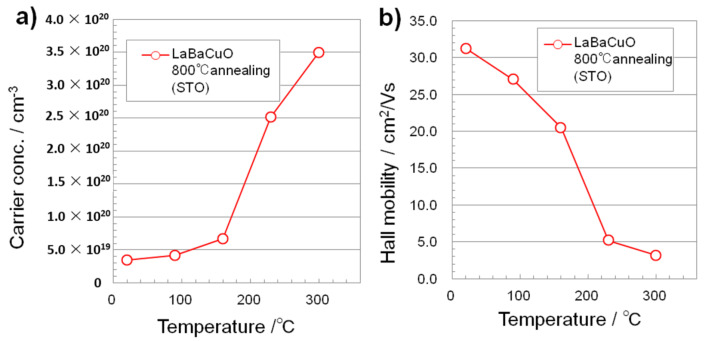
(**a**) Hall carrier concentration and (**b**) Hall mobility of La_4_BaCu_5_O_13+δ_ thin film on the substrate of SrTiO_3_ measured from room temperature to 300 °C. A single carrier model is assumed.

**Table 1 materials-14-02685-t001:** Comparison of the transport properties at 25 °C of the metals and LBCO materials of bulk and films (* data at 400 K, ** data at 746 K, ^†^ this study). TCR stands for temperature coefficient of resistance.

Material		Resistivity, ρ	TCR	Seebeck Coefficient, α	Bulk or Film Type [Ref.]
		(10^−6^ Ωcm)	(10^−3^/K)	(μV/K)	
Metal	Pt	10.6	3.9	−4.9	[17]
	Cupper	1.7	4.33	1.9	[18]
	Constantan	50	−0.03	−38	[18]
Perovskite cuprates	La_4_BaCu_5_O_13_	1000 *	0.24	4.0	Bulk [12]
		500	1.79	8.0	PLD film [13]
		5025 **	-	−510 **	Sputter thin film/Si ^†^
		1055 **	0.39	49 **	Sputter thick film/Si ^†^
		4115	−0.02	38	Sputter film/SrTiO_3_ ^†^
	La_4_BaCu_5−x_Co_x_O_13_	420 *	0.20	11	Bulk [12]
		1100	−0.22	13 *	PLD film [14]
		2473	0.10	13	Sputter film/SrTiO_3_ ^†^

## Data Availability

Data sharing is not applicable.

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
