# Peer review of "High Temperature Electrical Properties of Co-Substituted La4BaCu5O13+δ Thin Films Fabricated by Sputtering Method"

_materials, 2021, doi:10.3390/ma14102685_

Round 1
Reviewer 1 Report
The manuscript submitted for review concerns on electric and thermoelectric properties of La4BaCu5O13 thin films fabricated by a sputtering method. Generally, the article is thoughtfully written and includes all required information about the methods. The results presented by the Authors are of good quality and, in my opinion, are feasible for publications in the Materials Journal, however, I have some crucial comments, that make my decision to recommend accepting this publication after revision.
Comments:
- Figure 1 – it is not a crucial comment but I think that the axis orientation should be added to this picture. Also, the legend in form of single atoms with appropriate assignments could improve the clarity of this figure.
- The Authors sputtered LBCO films on the Si substrate. In the manuscript text, it is clear that the type of substrate significantly influences the properties of the thin films. To the best of my knowledge the orientation of the Si substrate also can influence the electrical properties of the thin films, thus the information on this Si orientation as well as some comments concerning this orientation on the LBCO properties should be also included in the manuscript (4.1. and 4.2. section)
- The XRD patterns – in figure 3a the LBCO thin films annealed in 1000 C degree decompose to few other phases. In my opinion, the additional phases that occurred in this temperature should be also identified and assigned in this figure. Moreover, LBCO annealed at 800 C degree also possesses additional reflection at about 31 2Q - is it the reflection from the LBCO phase or not?
- The XRD patterns – the Authors state about the orientation of the LBCO crystallites on the basis of the main reflection at about 33-34 2 I think that full Rietveld analysis should be performed for patterns presented in figure 3a (patterns 1 and 2), including preferred crystallites orientation, and Goodness of Fit (GoF) and Weighted R profile parameters.
- Figure 4 – there are no assignments, which points are assigned to conductivity/Seebeck coefficient in the legend, thus the figure is not clear for me.
- The most important comment – figure 4 – why this figure is cut? where are the points obtained for LBCO thin films in the low-temperature ranges? if the presented tendency of the red points is correct, the Seebeck coefficient in the low temperatures is about – 1000 μVK-1 – I think that such value is impossible to obtain for these materials. It must be explained! Here, the relation of the previously mentioned Si orientation should be also taken into account.
- Replacing Cu ions by Co ones, significant changes in oxygen vacancy concentration are expected. I believe that the Authors should address this substitution by analyzing the changes in Seebeck coefficient values.
- There are some editorial mistakes:
- Double spaces: line 73, line 265
- Something wrong with the font in line 75 (point “2.1”)
- Different fonts and poor quality of the fonts in figure 3, different fonts (bold or not) in figure 4 and 5
Author Response
Thank you for your valuable comments. I have revised and added the parts of manuscript referring your advice. I would like to answer your comments in detail as attached file:
Comments from reviewers
followed sentences: responses from authors
- Figure 1 – it is not a crucial comment but I think that the axis orientation should be added to this picture. Also, the legend in form of single atoms with appropriate assignments could improve the clarity of this figure.
Thank you for your comments. The figure is modified as commented.
- The Authors sputtered LBCO films on the Si substrate. In the manuscript text, it is clear that the type of substrate significantly influences the properties of the thin films. To the best of my knowledge the orientation of the Si substrate also can influence the electrical properties of the thin films, thus the information on this Si orientation as well as some comments concerning this orientation on the LBCO properties should be also included in the manuscript (4.1. and 4.2. section)
As commented, the orientation of STO substrate and Si is important information of the film growth. In the case of Si, as the film grow on the SiO2 surface the lattice matching is not critical. However, in the case of the oxide crystal substrate, lattice matching would be critical. If a LBCO film grows on the SrTiO3 lattice without rotation, the lattice misfit between the a-axis of La4BaCu5O13+δ (≈8.65 Å) and the length of a two-unit cell of SrTiO2 (7.82 Å) would be 10.6%. On the other hand, in the case with 26.6° rotation, lattice misfit becomes −1.1%, as discussed in our previous report (Tsuruta et al., (2016)). In the section of 3.1 and 4.2, we have added related discussions.
- The XRD patterns – in figure 3a the LBCO thin films annealed in 1000 C degree decompose to few other phases. In my opinion, the additional phases that occurred in this temperature should be also identified and assigned in this figure. Moreover, LBCO annealed at 800 C degree also possesses additional reflection at about 31 2Q - is it the reflection from the LBCO phase or not?
Regarding the identification of heterogeneous phases at 1000°C, we have investigated the peaks using Pearson's Crystal Data, but did not hit with the likely oxides of the La-Cu / Cu-Ba system. I think there is a possibility that Si is diffused to form a compound. We regard that the 31 ° peak at 800°C seems to be 300 planes and 201 planes, but the related peaks (600 and 402) are originally weak and the peak shape is sharper than others, so it is difficult to judge. We think that there is no choice but to judge by the intensity ratio with other peaks. We have added the assignment of La415 peaks in the revised Fig. 3.
- The XRD patterns – the Authors state about the orientation of the LBCO crystallites on the basis of the main reflection at about 33-34 2 I think that full Rietveld analysis should be performed for patterns presented in figure 3a (patterns 1 and 2), including preferred crystallites orientation, and Goodness
of Fit (GoF) and Weighted R profile parameters.
It may be good to carry out Rietveld analysis as recommended, but we could not perform this well, because the peak intensity ratio was random and poor to work on this analysis.
- Figure 4 – there are no assignments, which points are assigned to conductivity/Seebeck coefficient in the legend, thus the figure is not clear for me.
Thank you for your comments. The Figure 4 is modified as commented, and the Figure 5 is also revised.
- The most important comment – figure 4 – why this figure is cut? where are the points obtained for LBCO thin films in the low-temperature ranges? if the presented tendency of the red points is correct, the Seebeck coefficient in the low temperatures is about – 1000 μVK – I think that such value
is impossible to obtain for these materials. It must be explained! Here, the relation of the previously mentioned Si orientation should be also taken into account.
As pointed out by reviewer, high negative value of Seebeck coefficient over 1 mV/K and credibility of the measurement becomes doubtful, even though the materials have wide band gap. That is the reason we have cut the figure. The relation of the Si orientation was not critical because the film properties are too poor to discuss the effect of the lattice matching between film and substrate.
- Replacing Cu ions by Co ones, significant changes in oxygen vacancy concentration are expected. I believe that the Authors should address this substitution by analyzing.
The effect of Co substitution is discussed in detail in previous reports (Mori et al., 2015), and the concentration of the oxygen vacancy was also discussed using bulk samples, so that we referred this paper in the text.
Reviewer 2 Report
The manuscript entitled: “High temperature electrical properties of La4BaCu5O13+δ thin films fabrication by sputtering method” submitted to Materials- MDPI as a Article describes preparation of high temperature conductive mixed oxide as perovskite cuprate sputtered at SiO/Si or SrTiO3 single crystal substrate. Thermal process was applied to form required crystalline structure. In following step a substitution of Cu atoms with Co were done to improve electrical conductivity. The new material was characterized by XRD, AFM, SEM, commercial electrical conductivity and Seebeck coefficient measurement system.
After careful revision of the manuscript I do not recommend this work to be published in Materials- MDPI in its current form. The manuscript requires a significant improvement.
The list of issues than need to be addressed:
- Introduction: Authors mention a vast number of application for conductive oxides however provided only one reference? This fact leads to doubt such statement. I would suggest adding more references on the subject at least one for each application.
- Introduction: it would be clearer to identity which are the physical parameters, i.e. specific conductivity at what temperature are required for specific application.
- Introduction: it is not highlighted the advantages of used preparation technique (sputtering) over described in literature methods. Authors should highlight the advantages, like reproducibility of the process, scalability etc. Other advantages/ novelty should be also stated in this paragraph. If not the manuscript gives an impression of low or no scientific impact.
- Results and discussion: the AFM analysis was not fully described by authors. It would be easier to show differences or no differences in different samples comparing the roughness factors, grain size etc. Did authors perform AFM analysis after annealing the sample. The current description does not give clear evidence.
- Figures: Figure 4 and 5 should be improved due overlapping I would suggest scale modification to improve clarity.
- Conclusion: In my opinion the conclusion should be rewritten to clearly highlight all advantages of prepared material that might be of use by industry in future.
Author Response
Thank you for your valuable comments. I have revised and added the parts of manuscript referring your advice. I would like to answer your comments in detail as follow:
Comments from reviewers
followed sentences: responses from authors
- Introduction: Authors mention a vast number of application for conductive oxides however provided only one reference? This fact leads to doubt such statement. I would suggest adding more references on the subject at least one for each application.
As commented by reviewer, every application of the oxide is referred by additional references.
- Introduction: it would be clearer to identity which are the physical parameters, i.e. specific conductivity at what temperature are required for specific application.
As commented by reviewer, the temperature domain and conductivity requirements are added in the introduction as commented.
- Introduction: it is not highlighted the advantages of used preparation technique (sputtering) over described in literature methods. Authors should highlight the advantages, like reproducibility of the process, scalability etc. Other advantages/ novelty should be also stated in this paragraph. If not the manuscript gives an impression of low or no scientific impact.
Thank you for the advices. The highlighted advantages of sputtering the advantages, like reproducibility of the process, scalability etc. are addressed in the revised text.
- Results and discussion: the AFM analysis was not fully described by authors. It would be easier to show differences or no differences in different samples comparing the roughness factors, grain size etc. Did authors perform AFM analysis after annealing the sample. The current description does not give clear evidence.
The grain size was around several hundred nanometers for the films on SiO/Si substrate, and it became smaller than hundred nanometers for the film on SrTiO3. The agglomerate grain structure of irregular shapes, 200–300 nm wide was formed on the surface of the films on SiO/Si shows hillock. Because of these hillocks it was difficult to analysis the surface roughness quantitively in nanometer level. In the case of Co substituted LBCO film, no distinct change was found but the agglomerate grain structure was unclear on this sample.
- Figures: Figure 4 and 5 should be improved due overlapping
I would suggest scale modification to improve clarity.
We appreciated the comment. The scale is modified as recommended.
- Conclusion: In my opinion the conclusion should be rewritten to clearly highlight all advantages of prepared material that might be of use by industry in future.
As commented by reviewer, the content of the text, the introduction and in the discussion, has been rewritten, emphasizing the advantages of the process and properties of the oxides for the applications.
Reviewer 3 Report
The study did not reveal any specific improvement that haven’t been mentioned before in the literature. It is very bad explained. Reader could be totally become in mess while reading. The authors coamre many parameters at the dame time, refers to bulk (undoped) LBCO or doped, either substrates, thinness, layer or pellets etc. The manuscript should be carefully rewritten pressing attention to incongruity between comparisons of samples. The title should be Co-doped LBCO as it is the one that is studed in the paper and compared to the LBCO. The papers is very difficult to be followed during reading as the author is changing parameters and samples constantly, making comparison that cannot be done as more than one parameter is different. This affect to the overall discussion.
- I hardly understand what exactly is the novelty of the paper. Please state the novelty of the research.
- The T cited in the literature are quite lower than the ones you are annealing. Why did you anneal at higher T?
- How did you know the exact chemcial composition of the sputtered layer?
- XRD results: Fig 3a. LBCO pdf card number is missing in order to check if the peaks correspond exactly to LBCO as the author states. The SiO2 peaks coming from the substrates are not indexed. How the reader can know if the multiples peaks correspond only to LBCO or to the substrate? Please index it and mention in the text if the SiO/Si substrate is amorphous and in the background. Bad graphic quality of the difractogram. What 2theta/w axe means? Should be only 2theta.
- 3 b) Aging the x-axe is wrong (2theta/theta). What does it indexed in the red difractogram with black text? Which sample is represented in Fig 3b? In the text is stated that is the one at 800C, but must be written in the Figure explanation. The LBCO layer crystallinity is very low comparing to the one sputtered on Si. How can you explain that?
Moreover, the peak position you stated to correspond to LBCO in Fig 3a) does not correspond to the one you stated at Fig. 3b. Can you explain that?
- Electrical properties:
In the experimental part is written that the electrical measurements are taken from 50 to 600C, but in the text and in the figures appears up to 480. Why?
“A thicker film of 1m was prepared 150 and moderate thermal annealing condition of lower temperature of 600°C for 12 h was applied to improve the conductivity of the film, resulting in better electrical conduction. ”
Why this experiment is not mention in the experimental part? The thinness is crucial for electrical properties. How did you determine the layer thinness? By SEM? Can you show a cross section of the images?Why the leyres with different thickness are not on the XRD results discussion?
Fig 4 needs more detailed description from that: “High temperature thermoelectric properties of the LBCO films.” It is not enough as it is unknown the thickness of Si and STO layers. Could you resume the results from Fig. 4?
“The temperature coefficient of resistance, TCR, of the LBCO film became small, as the quality of the film was improved.” What do you mean by “improvement”? How did you improve the quality of the films? Did you referre to change thickness, annealing T or soaking time? In the figure, you exposed mixed parameters.
Figure 5. High temperature thermoelectric properties of the LBCO films substituted Co. Why didn’t you explain the pellet samples if you compared it in the figure.
Fig 6.” The SEM observation of the film on SrTiO3 showed just a smooth surface and is not shown here. “This could be discussed in the text.
etc...
Author Response
Thank you for your valuable comments. I have revised and added the parts of manuscript referring your advice. I would like to answer your comments in detail as follow:
Comments from reviewers
Followed sentences: responses from authors
The study did not reveal any specific improvement that haven’t been mentioned before in the literature. It is very bad explained. Reader could be totally become in mess while reading. The authors coamre many parameters at the dame time, refers to bulk (undoped) LBCO or doped, either substrates, thinness, layer or pellets etc. The manuscript should be carefully rewritten pressing attention to incongruity between comparisons of
samples. The title should be Co-doped LBCO as it is the one that is studied in the paper and compared to the LBCO. The papers is very difficult to be followed during reading as the author is changing parameters and samples constantly, making comparison that cannot be done as more than one parameter is different. This affect to the overall discussion. I hardly understand what exactly is the novelty of the paper. Please state the novelty of the research.
As reviewer pointed out, the novelty of the paper would be the Co-substituted LBCO film and its electrical properties at high temperature. This study was triggered from the paper which reported a set of polycrystalline samples of the metallic copper oxide La4BaCu5xCoxO13 (0-x-0.35) with the interesting discussions on the two-carrier model and the transport properties explained by a band picture. In this oxide system, it is very interesting that the Seebeck sign and the carrier type expected by Hall voltage was not coincide, and we would explore whether this is also happed in high temperature.
The T cited in the literature are quite lower than the ones you are annealing. Why did you anneal at higher T? How did you know the exact chemical composition of the sputtered layer?
As commented, the annealing treatment temperature has a great effect on the characteristics of the thin film. Since we studied different process from the literature, the annealing temperature 600, 800, 1000 C was tested on Si substate. The annealing temperature in this study was higher than those reported, and this difference seems related to the deposition process.
The exact composition is unknown. In the XRD results, no CoO related phase detected so that we regarded the substitution was occurred after high temperature annealing. It was reported that a slight decrease in the a axis length (Mori et al., 2015), but is was unable to be determined from the x-ray diffraction patterns in this study. We discussed in the revised text.
XRD results:
Fig 3a. LBCO pdf card number is missing in order to check if the peaks correspond exactly to LBCO as the author states. The SiO2 peaks coming from the substrates are not indexed.
How the reader can know if the multiples peaks correspond only to LBCO or to the substrate? Please index it and mention in the text if the SiO/Si substrate is amorphous and in the background.
Bad graphic quality of the difractogram. What 2theta/w axe means? Should be only 2theta.
Considering that the peak of SiO2 or Si does not appear in the pattern of at least in samples of 600°C and 800°C, we could not distinguish the background peaks and SiO-originated peaks. Regarding the case of 1000℃, by the same reason, we could not make any index.
For the unit of horizontal axis, we have modified as recommended. In the case of a thin film, to be precise, it should be as 2 theta / ω when the substrate is surfaced in a specific direction. It also may be described as 2 theta / theta, however, it depends on the device-related measurement method.
3 b) Aging the x-axe is wrong (2theta/theta). What does it indexed in the red difractogram with black text? Which sample is represented in Fig 3b? In the text is stated that is the one at 800C, but must be written in the Figure explanation. The LBCO layer crystallinity is very low comparing to the one sputtered on Si. How can you explain that?
As reviewer commented, the horizontal axis of Figure 3 is modified, and the peak indexes are added. The crystallinity of the film was not worse than that of the film on the Si substrate, but the relative peak intensity of the single crystal STO substrate was very strong so that the peak of LBCO looked relatively weak. If the measurement had been carried out in exactly same way, we could have compared these more closely.
Moreover, the peak position you stated to correspond to LBCO in Fig 3a) does not correspond to the one you stated at Fig. 3b.
Can you explain that?
For the Peak positions,
120 > because the peak of the 001 plane of STO or the BG of the 001 peak is buried in the background of Cu Ka line cannot be cut completely in the case of a single crystal with a Ni filter (SmartLab, Rigaku, Tokyo, Japan).
131> the assignment is added.
240, 241> is buried in the peak of the 002 plane of STO
121, 341, and others > I confirmed again that they correspond to the LBCO.
Electrical properties:
In the experimental part is written that the electrical measurements are taken from 50 to 600C, but in the text and in the figures appears up to 480. Why?“A thicker film of 1mm was prepared 150 and moderate thermal annealing condition of lower temperature of 600°C for 12 h was applied to improve the conductivity of the film, resulting in better electrical conduction. ”
Why this experiment is not mention in the experimental part?
The thinness is crucial for electrical properties. How did you determine the layer thinness? By SEM? Can you show a cross section of the images? Why the layers with different thickness are not on the XRD results discussion?
The deposition rate was checked by a surface profilometer to be 6 nm/min, and the film thickness was controlled to 500 nm, as described in the experimental. We have checked cross sectional SEM and confirmed this is reasonable, but the image was not clear to be checked quantitatively. The change of thickness was not followed the annealing temperature variation, so that the results was not shown.
Fig 4 needs more detailed description from that: “High temperature thermoelectric properties of the LBCO films.” It is not enough as it is unknown the thickness of Si and STO layers. Could you resume the results from Fig. 4? “The temperature coefficient of resistance, TCR, of the LBCO film became small, as the quality of the film was improved.”What do you mean by “improvement”? How did you improve the quality of the films? Did you referre to change thickness, annealing T or soaking time? In the figure, you exposed mixed parameters.
As commented, the thickness of Si and STO layer are addressed in the revised text. For the high temperature thermoelectric properties of the LBCO films, we consider that the thickness of the substrate cannot effect the electrical properties because these are insulating substrate or insulating layer covered.
Figure 5. High temperature thermoelectric properties of the LBCO films substituted Co. Why didn’t you explain the pellet samples if you compared it in the figure.
In the text, we have discussed the effect of the Co substitution, and the pellet sample itself is a Co3O4, which is used in the film deposition process of Co substitution for Cu.
Fig 6.” The SEM observation of the film on SrTiO3 showed just a smooth surface and is not shown here. “This could be discussed in the text.
The text is modified and discussion on this is added as commented.
Round 2
Reviewer 1 Report
I accept all the authors' responses to my comments, except two of them, listed below:
1. The XRD patterns - the Authors reply "Regarding the identification of heterogeneous phases at 1000°C, we have investigated the peaks using Pearson's Crystal Data, but did not hit with the likely oxides of the La-Cu / Cu-Ba system. I think there is a possibility that Si is diffused to form a compound. We regard that the 31 ° peak at 800°C seems to be 300 planes and 201 planes, but the related peaks (600 and 402) are originally weak and the peak shape is sharper than others, so it is difficult to judge. We think that there is no choice but to judge by the intensity ratio with other peaks. We have added the assignment of La415 peaks in the revised Fig. 3."
Well, it must be explained in the manuscript text somehow. I would also insist to assign the reflections from other than LBCO phases at the figure, e.g. in form of dotes or other symbol.
2. The results of Seebeck coefficient presented in figure 4 - the Authors reply "As pointed out by reviewer, high negative value of Seebeck coefficient over 1 mV/K and credibility of the measurement becomes doubtful, even though the materials have wide band gap. That is the reason we have cut the figure."
If the measurement of the sample in the low-temperature range becomes doubtful, all other results become doubtful. It must be explained or comment on in the main text somehow. Is this a problem with the apparatus, some artificial effects, or any other?
The second comment still remains a critical remark, which influences my decision to revise this manuscript again.
Reviewer 2 Report
The manuscript entitled: “High temperature electrical properties of La4BaCu5O13+δ thin films fabrication by sputtering method” submitted to Materials- MDPI as an Article describes preparation of high temperature conductive mixed oxide as perovskite cuprate sputtered at SiO/Si or SrTiO3 single crystal substrate. Thermal process was applied to form required crystalline structure. In following step a substitution of Cu atoms with Co were done to improve electrical conductivity. The new material was characterized by XRD, AFM, SEM, commercial electrical conductivity and Seebeck coefficient measurement system.
After careful revision of the revised version of the manuscript I do recommend this work to be published in Materials- MDPI after major revision. The manuscript requires major linguistic revision, specially in the added sections of the manuscript. Authors addressed correctly my suggestions form the first round revision and done adequate correction to their work.
Reviewer 3 Report
My recommendations have not been stressed at all in the revised version.
